# Resin-Based Sealant with Bioactive Glass and Zwitterionic Material for Remineralisation and Multi-Species Biofilm Inhibition

**DOI:** 10.3390/nano10081581

**Published:** 2020-08-12

**Authors:** Myung-Jin Lee, Ji-Yeong Kim, Ji-Young Seo, Utkarsh Mangal, Jung-Yul Cha, Jae-Sung Kwon, Sung-Hwan Choi

**Affiliations:** 1Division of Health Science, Department of Dental Hygiene, Baekseok University, Cheonan 31065, Korea; dh.mjlee@gmail.com; 2Department of Orthodontics, Institute of Craniofacial Deformity, Yonsei University College of Dentistry, Seoul 03722, Korea; katekim826@yuhs.ac (J.-Y.K.); jyseo13@yuhs.ac (J.-Y.S.); utkmangal@yuhs.ac (U.M.); jungcha@yuhs.ac (J.-Y.C.); 3BK21 PLUS Project, Yonsei University College of Dentistry, Seoul 03722, Korea; 4Department and Research Institute of Dental Biomaterials and Bioengineering, Yonsei University College of Dentistry, Seoul 03722, Korea

**Keywords:** resin-based sealant, bioactive glass, zwitterion, remineralisation, bacterial adhesion, protein adsorption, caries inhibition

## Abstract

Since pits and fissures are the areas most commonly affected by caries due to their structural irregularity, bioactive resin-based sealant (RBS) may contribute to the prevention of secondary caries. This study aims to investigate the mechanical, physical, ion-release, enamel remineralisation, and antibacterial capabilities of the novel RBS with bioactive glass (BAG) and 2-methacryloyloxyethyl phosphorylcholine (MPC). For the synthesis, 12.5 wt% BAG and 3 wt% MPC were incorporated into RBS. The contact angle, flexural strength, water sorption, solubility, and viscosity were investigated. The release of multiple ions relating to enamel remineralisation was investigated. Further, the attachments of bovine serum albumin, brain heart infusion broth, and *Streptococcus mutans* on RBS were studied. Finally, the thickness and biomass of a human saliva-derived microsm biofilm model were analysed before aging, with static immersion aging and with thermocycling aging. In comparison to commercial RBS, BAG+MPC increased the wettability, water sorption, solubility, viscosity, and release of multiple ions, while the flexural strength did not significantly differ. Furthermore, RBS with MPC and BAG+MPC significantly reduced protein and bacteria adhesion and suppressed multi-species biofilm attachment regardless of the existence of aging and its type. The novel RBS has great potential to facilitate enamel remineralisation and suppress biofilm adhesion, which could prevent secondary dental caries.

## 1. Introduction

Dental caries are one of the most common oral diseases with its highest rate occurring on occlusal surfaces [1,2]. Caries-preventive approaches, such as topical fluoride application, plaque control, and dietary control, have generally reduced the prevalence of smooth surface caries [3,4]. However, the pits and fissures of permanent molars are vulnerable sites for caries lesions due to their complex morphologies, which are ideal for retaining bacteria and food residues [5,6,7]. To inhibit this problem, pit and fissure sealants have been widely used as a preventive strategy, especially in children aged between 6 and 16 [6]. Pit and fissure sealants are applied to provide a physical barrier and to manage early carious lesions [8]. Resin-based sealants (RBS) based on bisphenol A glycidyl methacrylate were found to the most effective material due to its aesthetics, proper flowability, and physical properties [9]. However, RBS tend to be more susceptible to accumulating plaque and biofilm than other materials [10,11]. Moreover, microleakage is one of its biggest drawbacks as it causes bacterial invasion and secondary caries [12]. Particularly, marginal caries are formed around the sealed sealant on the material–tooth interfaces due to either the partial loss of materials or the microleakage and gaps induced by polymerisation [13,14]. Furthermore, carious lesions around the sealant are one of the main causes of their failure [15,16]. 

Several studies have been focused on addressing the issue of marginal and secondary caries [10,17]. Bioactive glass (BAG) is a class of biomaterial normally based on amorphous silicate compounds [18,19]. Among these, 45S5 BAG is widely known for remineralising tooth structure [20]. Previous studies exhibited the significantly higher calcium and phosphate ion release and buffering capacity of BAG-containing dental materials, which promote remineralisation and suppress demineralisation [21]. However, BAG-containing dental materials have a weak antibacterial effect. 

Oral bacteria are attached to dental resins through a layer of adsorbed salivary proteins on the resin surface, which is a prerequisite for bacterial attachment and biofilm growth [4,22]. Accordingly, the protein-repellent property of resin helps in repelling bacteria attachment [23,24]. Recently, zwitterionic materials have attracted considerable attention due to their excellent protein-repellent, antifouling, and stability properties [25]. Thus, with their positive and negative charges, they have been recognised as effective antifouling materials [17,26]. Particularly, 2-methacryloyloxyethyl phosphorylcholine (MPC) exhibit excellent antifouling effect due to their hydrophilicity and strong water interaction [27,28]. Moreover, numerous studies have demonstrated the potential of MPC-containing dental materials in repelling proteins, bacteria, and biofilm [17,29].

Dental caries originate in tooth demineralisation by organic acids from the fermentation of dietary carbohydrates by oral bacteria [30]. There are two main approaches to prevent marginal or secondary caries—promote remineralisation and inhibit biofilm formation [2,10,30]. Accordingly, the ultimate goal of novel dental materials is to prevent marginal or secondary caries and develop RBS with enamel remineralisation and antibacterial capabilities [17]. Also, viscosity is an important factor in the application of sealants. However, to date, there is yet to be a study on the viscosity and benefits of RBS containing BAG and MPC in providing both remineralisation and antibacterial capabilities. 

Hence, this study aims to develop RBS incorporated with BAG and MPC and its remineralisation and antibacterial properties, while maintaining the original valuable properties of RBS. The null hypotheses state that RBS incorporating BAG and MPC would not result in significant changes in the (1) physical and mechanical properties including viscosity, (2) multiple ion release related to enamel remineralisation, and (3) antibacterial properties of commercial RBS. 

## 2. Material and Methods

### 2.1. Preparation of 45S5 BAG and MPC Powder

High-purity silicon dioxide (Junsei Chemical Co., Tokyo, Japan), sodium carbonate (Duksan Pure Chemicals Co., Ansan-city, Korea), calcium carbonate (Samchun Pure Chemicals Co., Pyeongtaek city, Korea), and phosphorus pentoxide (Sigma-Aldrich, Steinheim, Germany) powders weighed 45, 24.5, 24.5, and 6 wt%, respectively, were used in the preparation of 45S5 BAG. The powders were mixed and melted in a platinum crucible at 1400 °C for 1 h and quenched on a graphite plate mould at room temperature. The melt-derived 45S5 BAG was ground using a fast mill (Ceramic Instruments, Sassuolo, Italy). The ground powder was filtered through a 500 mesh sieve to obtain fine particles less than 25 µm in size [21]. Commercially available MPC powder (Sigma–Aldrich, St. Louis, MO, USA) was used as the zwitterionic material. Based on previous studies [21], 12.5 wt% BAG powder was incorporated into the glass filler. Subsequently, 3 wt% MPC powder, which was found to be the most effective quantity [17,26,28], was mixed into RBS.

### 2.2. Fabrication of Novel RBS

The resin matrix of bisphenol A glycerolatedimethacrylate (Sigma-Aldrich, Steinheim, Germany) and triethylene glycol dimethacrylate (Sigma-Aldrich, Steinheim, Germany) with 1:1 mass ratio was mixed with 0.3% camphorquinone (Sigma-Aldrich, Steinheim, Germany) and 0.6% 2-(dimethylamino)ethyl methacrylate (Sigma-Aldrich, Steinheim, Germany). For mechanical reinforcement, silanised dental glass filler (180 ± 30 nm; NanoFine NF180; Schott, Landshut, Germany) was used as the conventional glass filler in the composite resin. The silanised dental glass filler and 12.5 wt% BAG were added into the resin matrix and mixed with a magnetic stirrer in a dark environment for 24 h to obtain a homogeneous mixture [21]. Through hand mixing, 3 wt% MPC powder was mixed into the liquid-like state of RBS. The samples were vortexed and mixed using a high-speed mixer (SpeedMixer, Hauschild, Hamm, Germany) [17] and polymerised using a LED light-curing unit (Elipar S10; 3M ESPE Co., Seefeld, Germany). Commercially available RBS (Helioseal, Ivoclar Vivadent, Liechtenstein) was used as the control. Accordingly, these five groups were tested: (1)Commercial control (‘CC’)(2)Experimental control, 50 wt% resin matrix + 50 wt% glass filler (‘EC’)(3)48.5 wt% resin matrix + 48.5 wt% glass filler +3 wt% MPC (‘MPC)(4)50 wt% resin matrix + 37.5 wt% glass filler + 12.5 wt% BAG (‘BAG’)(5)48.5 wt% resin matrix + 36.38 wt% glass filler + 12.12 wt% BAG + 3 wt% MPC (‘MPC+BAG’)

### 2.3. Wettability

The wettability of the samples was measured in accordance with previous studies [17,28]. Distilled water was chosen as the reference liquid. The samples were placed in a mould with 15 mm diameter and 2 mm thickness to form disc-shaped samples. They were then polymerised using a LED light-curing unit (Elipar S10; 3 M ESPE Co., Seefeld, Germany). The static contact angle 10 s after the drop of 3 µL distilled water on the disc surface was measured using a video contact angle goniometer (SmartDrop, Femtobiomed Inc., Gyeonggi-do, Korea). 

### 2.4. Flexural Strength

The mechanical properties of the samples were measured using a three-point flexural strength method in accordance with ISO 4049 (2019). Bar-shaped samples were prepared without air bubbles or voids using a 25 × 2 × 2 mm mould. All samples were polymerised using a LED light-curing unit (Elipar S10; 3M ESPE Co., Seefeld, Germany) and then stored in distilled water at 37 ± 1 °C for 24 h. The maximum loads were measured by a universal testing machine (Instron 5942, Istron, Norwood, MA, USA) at a crosshead speed of 1 mm/min. The flexural strength was then calculated according to S = 3 Fl/(2 bh^2^), where F is the maximum fracture load, and l, b, and h are the span, width, and thickness of the specimen, respectively.

### 2.5. Water Sorption and Solubility

The water sorption and solubility test were performed according to ISO 4049 (2019). Each material was placed in a mould with 15 mm diameter and 1 mm height. The average diameter and thickness of the samples were calculated by measuring two diameters along its length and four equally spaced points of the circumference, respectively. The values obtained were then used to calculate the volume (*V*) of all samples (in 0.01 mm^3^). Each sample weight was measured by an analytical balance (accurate to 0.01 mg) (XS105, Mettlertoledo AG, Greifensee, Switzerland) with a reproducibility of 0.1 mg until a constant mass (*m*_1_) was obtained. All samples were immersed in distilled water and placed in a water bath maintained at 37 °C for 7 days. Subsequently, the samples were blotted until there was no visible moisture, shaken in the air for 15 s, and weighed to determine the final mass (*m*_2_). Finally, the samples were placed in a desiccator and weighed daily until a constant dry mass (*m*_3_) was obtained. Water sorption (Wsp) in g/mm3 was calculated by Equation (1) and water solubility (Wsl) was calculated by Equation (2).
(1)$$W_{sp}=\frac{m_{2}-m_{3}}{V}\;$$
(2)$$W_{sl}=\frac{m_{1}-m_{3}}{V}\;$$

### 2.6. Viscosity

To evaluate the flow properties of the sealants, the viscosity was measured using a stress-controlled rheometer (MCR 702, Anton Paar GmbH, Filderstadt, Germany) at a constant temperature of 25 °C using a Peltier temperature device. Due to the low viscosity of pit and fissure sealants, a parallel plate rheometer module with 50 mm diameter (PP50) and 1 mm plate spacing was chosen. The data were collected at shear rates of 0.1–800 s^−1^. For the viscosity test, the sample groups are combined as follows: resin matrix without glass filler (RM) and resin matrix with MPC 3 wt% and without glass filler (MPC 3%). 

### 2.7. Ion Release

The material filled a mould of 10 mm diameter and 2 mm height to prepare the samples. They were then stored in 5 mL distilled water at 37 °C. After 24 h, samples containing eluted ions were collected and the concentration of each ion was measured. The elemental analysis of Na^+^, Ba^2+^, Ca^2+^, Al^3+^, SiO_3_^2−^, BO_3_^3−^, and PO_4_^3−^ ions released from the discs was performed using an inductively coupled plasma optical emission spectrometer (Agilent 5100, Agilent Technologies, Santa Clara, CA, USA).

### 2.8. Protein Adsorption

The protein adsorption was investigated according to a previously established method [17,28]. Each material was placed in a mould with 15 mm diameter and 2 mm thickness to form disc-shaped samples. All samples were immersed into fresh phosphate-buffered saline (PBS; Gibco, Grand Island, NY, USA) for 1 h at room temperature and immediately immersed into a protein solution (2 mg/mL PBS) of either a bovine serum albumin (BSA; Pierce Biotechnology, Rockford, IL, USA) or brain heart infusion (BHI; Difco, Sparks, MD, USA) broth (volume of 100 μL). After incubation at 37 °C for 1 h, the samples were gently rinsed twice with fresh PBS. After 4 h of incubation under sterile humid conditions at 37 °C, any protein that did not adhere to the samples was removed by washing them twice with PBS. The amount of protein adhered to samples was measured using 200 μL micro-bicinchoninic acid (Micro BCATM Protein Assay Kit; Pierce Biotechnology) followed by incubation at 37 °C for 30 min. The quantitative analysis of the adsorbed proteins on the surfaces was calculated following the absorbance measurement at 562 nm using a micro-plate reader (Epoch, BioTek Instruments, Winooski, VT, USA). 

### 2.9. Bacterial Attachment and Viability

Bacterial analyses were performed using *Streptococcus mutans* (ATCC 25175). *S. mutans* was cultured in BHI broth (Difco, Sparks, MD, USA) under aerobic conditions in an incubator at 37 °C. Following the preparation of disc-shaped specimens, 1 mL of bacterial suspension (1 × 10^8^ cells/mL) was placed on each disc in a 24-well plate and incubated at 37 °C for 24 h. After incubation, the samples were gently washed twice with PBS to remove any non-adherent bacteria [17].

For the microscopic examination, the bacteria attached to the samples were fixed with 2% glutaraldehyde-paraformaldehyde in 0.1 M PBS for at least 30 min at room temperature. The samples were post-fixed with 1% OsO4 dissolved in 0.1 M PBS for 2 h, dehydrated in an ascending gradual series of ethanol, treated with isoamyl acetate, and subjected to critical point drying (LEICA EM CPD300; Leica, Wien, Austria). Then, the discs were coated with 5 nm Pt using an ion coater (ACE600; Leica). They were then examined and photographed via field emission scanning electron microscopy (FE-SEM; Merin, Carl Zeiss, Oberkochen, Germany) at 2 kV. 

The bacterial viability was examined using water-soluble tetrazolium salt reagent (WST; EZ-CYTOX, Dogenbio, Seoul, Korea) following the manufacturer protocol [31]. WST assay comprised of sensitive colorimetric assays to determine bacterial viability and a yellow tetrazole to formazan. *S. mutans* was cultured in the same way as mentioned above. The samples were placed in the cultured bacterial solution and rinsed in fresh PBS (Gibco, Grand Island, NY, USA) after 24 h to remove loose bacteria. Subsequently, the samples were placed in a 24-well plate (SPL, Pochein-Si, Gyeonggi-Do, Korea) with 1 mL PBS and sonicated (SH-2100; Saehan Ultrasonic, Seoul, Korea) for 5 min. Subsequently, 100 µL bacterial solution was added to 10 µL WST reagent in a 96-well plate and incubated for 4 h. The optical density of WST was measured at 450 nm with a microplate spectrophotometer (BioTek, Winooski, VT, USA).

### 2.10. Saliva-Derived Biofilm Model and Biomass Measurement

Since the release of ions from RBS in the oral cavity could decrease bioactivity over time, biofilm experiments were carried out under three conditions: before aging, with static immersion aging, and with thermocycling aging [32,33]. For static immersion aging, the samples were prepared by storing them in distilled water for 7 days. The immersion temperatures ranged from 5 °C to 55 °C. For thermocycling aging, the samples were prepared with thermocycling equipment (Thermal Cyclic Tester, R&B Inc., Daejeon, Korea) set at 45 s dips to 5 s transfer time for 850 cycles, corresponding to 1 month. 

Human saliva has many advantages in maintaining the complexity and heterogeneity of dental plaque in vivo and is ideal for growing plaque microcosm biofilms in vitro [34]. Following the previous study [17], human saliva was collected from healthy adult donors without active caries or periodontal disease and who had not taken antibiotics within the past 3 months, following the procedures approved by the institutional review board of the Yonsei University Dental Hospital (Seoul, Republic of Korea) (2- 2019-0049). The donors did not brush their teeth for 24 h and did not consume any foods or drinks for at least 2 h before donating saliva. The saliva was then collected from six individuals and mixed in equal proportions, diluted in sterile glycerol to a concentration of 30%, and stored at −80 °C to be used as the biofilm model. 

The biofilm model was cultured in McBain medium supplemented with mucin (type II, porcine, gastric) (2.5 g/L), bacteriological peptone (2.0 g/L), tryptone (2.0 g/L), yeast extract (1.0 g/L), NaCl (0.35 g/L), KCl (0.2 g/L), CaCl_2_ (0.2 g/L), cysteine hydrochloride (0.1 g/L), haemin (0.001 g/L), and vitamin K1 (0.0002 g/L) at 37 °C for 24 h. From the cultured medium, 1.5 mL bacterial solution was placed on the specimen. After 8 h, 16 h, and 24 h of incubation, respectively, 1.5 mL bacterial solution was added on the specimen. The biofilms were grown for 48 h. Finally, the viability of the adhered bacteria was examined by staining using a live/dead bacterial viability kit (Molecular Probes, Eugene, OR, USA) according to the manufacturer protocols. From the kit, equal volumes of Syto 9 dye and propidium iodide were mixed thoroughly to stain live and dead bacteria, respectively. Subsequently, 3 µL mixture was added to 1 mL bacterial suspension. After storing in the dark at room temperature for 15 min, the stained samples were observed under a confocal laser microscope (CLSM, LSM880, Carl Zeiss, Thornwood, NY, USA), where live and dead bacteria are coloured green and red, respectively. The biofilm was then visualised at three randomly chosen positions using CLSM (LSM880, Carl Zeiss, Thornwood, NY, USA). The axially stacked biofilm images were captured and the biofilm thickness was calculated using the software (Zen, Carl Zeiss, Thornwood, NY, USA). Additionally, the COMSTAT plug-in (Technical University of Denmark, Kongens Lyngby, Denmark) and the ImageJ (NIH, Bathesda, MA, USA) software were used to determine the average biomass.

### 2.11. Statistical Analysis

All statistical analyses were conducted with IBM SPSS software, version 23.0 (IBM Korea Inc., Seoul, Korea) for Windows. The results obtained for the control and experimental groups were analysed by one-way analysis of the variance (ANOVA) followed by the Tukey’s test. 

## 3. Results

### 3.1. Physical and Mechanical Properties

The water contact angles are shown in Figure 1A. There are no significant differences between the water contact angles of CC (77.36 ± 8.95°) and EC (84.63 ± 12.14°). Meanwhile, MPC, BAG, and MPC+BAG had significantly lower contact angles than EC (*p* < 0.001). MPC+BAG has the smallest contact angle (65.35 ± 6.78°) among all groups. From Figure 1B, the flexural strength of MPC (54.12 ± 2.55 MPa) significantly decreased from that of CC (76.02 ± 10.96 MPa) and EC (69.95 ± 5.26 MPa) (*p* = 0.005). On the other hand, the flexural strength of BAG (69.35 ± 7.18 MPa) and MPC+BAG (61.32 ± 6.16 MPa) did not significantly different from that of the control group. 

The water sorption and solubility results are shown in Figure 1C,D, respectively. There was no statistically significant difference between the water sorption and solubility of MPC and BAG. However, MPC+BAG exhibits the largest values for both the water sorption and solubility (*p* < 0.001), indicating the amplified water absorption and solubility when MPC and BAG are mixed.

### 3.2. Viscosity

The viscosities of the different groups are shown in Figure 2. The viscosity of MPC and MPC+BAG is higher than that of EC and BAG. MPC 3% has a higher viscosity than RM, which indicates the increased viscosity as MPC is contained.

### 3.3. Ion-Releasing Properties

The ion-releasing properties of the samples in terms of leached Na^+^, Ba^2+^, Ca^2+^, Al^3+^, SiO_3_^2−^, BO_3_^3−^, and PO_4_^3−^ ions are shown in Table 1. In comparison to BAG, the release of the ions, except for the ion of Ba^2+^, increased for MPC+BAG (*p* < 0.001). In terms of the Na^+^ and Al^3+^ ions, there were no significant differences between BAG and MPC+BAG, unlike that of Ca^2+^, SiO_3_^2−^, BO_3_^3−^, and PO_4_^3−^ ions. Particularly, the amount of PO_4_^3−^ ions increased to more than 100-fold from those of MPC and BAG. 

### 3.4. Protein Adsorption

Figure 3 shows the protein adsorption of the different groups. There is no significant difference in the amount of adsorbed BSA from BAG and that of the control groups, while those of MPC and MPC+BAG are significantly lower (*p* < 0.001). Figure 3B shows the amount of proteins adsorbed from BHI medium. Both graphs exhibit a similar trend. Notably, MPC+BAG has the lowest adsorption (*p* < 0.001).

### 3.5. Bacterial Attachment and Viability

Figure 4 shows the representative FE-SEM images and WST analysis of attached *S. mutans* on the RBS surface. The images clearly show that less *S. mutans* were attached to the surface of MPC and BAG+MPC than on those of CC, EC, and BAG. This result was further confirmed by the quantitative WST analysis, which revealed the lowest bacterial adhesion on MPC and MPC+BAG surfaces (*p* = 0.016). Moreover, there were no significant differences between the bacteria adhesion of CC, EC, and BAG. Overall, the WST results are consistent with those observed by FE-SEM. 

### 3.6. Biofilm Thickness and Biomass

Figure 5 shows the images, biofilm thickness, and biomass of the biofilms for different groups. Before aging, there is a distinct difference between the biofilms of MPC and MPC+BAG, and those of other groups. Particularly, MPC and MPC+BAG have thinner biofilms, which was confirmed by the software calculation of the thickness (*p* = 0.001, Figure 5B). Furthermore, the biomass of the biofilm was significantly reduced in MPC and MPC+BAG from those of the control groups and BAG (*p* = 0.001, Figure 5C). The additional analyses performed under different aging conditions showed similar trends to that before aging. In all groups except EC, there was no significant change despite the increase in biofilm thickness during static immersion and thermocycling aging. MPC+BAG has the smallest biofilm thickness regardless of the type of aging. With thermocycling aging, the biofilm biomass significantly increased in MPC and MPC+BAG, which are the smallest values.

## 4. Discussion

In preventive dentistry, RBS has been widely used as the most effective material in preventing caries within the pits and fissures of the occlusal surfaces [4,35,36]. However, RBS has several drawbacks, such as its easy exposure to bacterial attachment and growth in the oral cavity, which can cause secondary caries around RBS [8,35]. As long as RBS remains attached to the enamel, its protection continues effectively [37]. However, microleakage due to polymerisation eventually leads to marginal or secondary caries [1,14]. 

The basic mechanism of caries is the demineralisation of the tooth by the acid generated from a bacterial biofilm [10,28]. Thus, to prevent secondary caries, the critical strategies include inhibiting demineralisation and bacterial attachment [17,38], which cannot be performed by RBS [4]. To overcome this problem, preventive materials should be considered to develop RBS with the property to inhibit secondary caries [39]. 

BAG, a bioactive material introduced in 1971, has been used to promote enamel remineralisation [19]. The potential remineralising effect of BAG has attracted much attention due to its ability to release calcium and phosphate ions [18,20,40]. On the other hand, MPC is a zwitterionic material with a molecular structure optimised for antifouling [25,28,29]. MPC has been applied in various dental materials, including composite materials, varnish, and calcium silicate-based cement [17,26,27]. To date, there is yet to be a study on the twofold effects of RBS with BAG+MPC. Therefore, the present study firstly focused on investigating the remineralisation and antibacterial capacities of the synthesised RBS with BAG+MPC.

From the results, we partially accept the first null hypothesis stating that RBS with BAG+MPC would not exhibit significant differences in the physical and mechanical properties. However, in terms of wettability, there was significant difference compared to those of the control groups. Since bacteria tend to attach more on hydrophobic surfaces than on hydrophilic surfaces, an increase in the contact angle could affect the bacterial attachment [41,42]. Particularly, hydrophobic bacterial species, such as *S. mutans*, prefer hydrophobic surfaces and cannot easily attach to hydrophilic surfaces [43]. Furthermore, the major mechanism of the anti-adhesive effect against bacteria and its relation to the hydrophilic surface is consistent with our results [41,44]. 

In addition to wettability, sealants should also exhibit sufficient mechanical properties [17]. In the present study, incorporating BAG+MPC did not negatively influence the flexural strength of the sealant. Previous reports achieved a proper flexural strength of greater than 40 MPa for commercially available sealant [45], which was exceeded for BAG+MPC, thereby confirming its clinically sufficient mechanical strength. 

Water sorption and solubility are related to the mechanical and chemical properties of sealants in an aqueous environment [17,21]. In this study, incorporating BAG+MPC yielded a significant difference in the water sorption and solubility. In Figure 1C,D, BAG+MPC showed high water sorption and solubility as a result of the hydrophilic properties of bioactive glass and zwitterionic materials [25,46]. As MPC is an ionic substance, it can absorb moisture into its core [27], which increases the amount of ion release, thereby increasing its water sorption and solubility. In addition, P_2_O_5_ of BAG has a strong moisture absorption property, which is associated with the increased water sorption and solubility [21,47]. Martignon et al. [7] clinically demonstrated the reliable retention of RBS with BAG on pits and fissures for a year. Although it did not exactly match the materials used in this study, it can be assumed that BAG+MPC with similar bioactive properties would have similar results. 

In the viscosity analysis, the viscosity of BAG+MPC groups increased from those of the control groups. Although high viscosity glass ionomer, as a pit and fissure sealant, does not flow easily, it can be placed by curing after compressing a small amount on the tooth surface, a method that has been used effectively for a long time. It is inferred that the viscosity increases as the phosphorus concentration rapidly increases [48]. As the amount of ion release increases with increasing water sorption and solubility, Ca and P ions would be essential in inhibiting demineralisation [40]. As long as RBS can be strongly attached to the tooth, it can prevent microleakage at the tooth–RBS margin [4]. Thus, the release of Ca and P ions from novel RBS is highly advantageous in acting as seed crystals to facilitate remineralisation for the microleakage at the interface between the material and tooth structure [47,49]. 

BAG is an inorganic compound of biocompatible materials that reacts rapidly in an aqueous environment with the releases of ions, such as Ca^2+^, Na^+^, and PO_4_^3−^, which could aid in the remineralisation of the tooth structure [21]. According to Shimazi et al. [50], BAG has a high ability to release ions due to its large surface and high properties due to weak ionic bonds. The easy release of calcium into the surrounding environment could inhibit demineralisation and promote remineralisation [19]. In MPC, the rapid release of PO_4_^3−^ ions seems to play a role in the acidity of surrounding solutions [17]. The structure of MPC with polarised phospholipid side chains in the outward direction of the polarity and non-polar tail region inward in the bilayer could interact with liquid [17]. In this study, BAG+MPC groups had Ca^2+^, SiO_3_^2−^, BO_3_^3−^, and PO_4_^3−^ ion releases that were significantly higher than those of the control groups. Moreover, bioactive materials, such as BAG or MPC, possess acid-neutralising capacities in a cariogenic environment, which could inhibit caries and enhance the remineralising capabilities in marginal gaps [20,21]. The results confirmed the bioactive properties of the novel RBS in the present study, which can provide important insights in suppressing demineralisation and promoting the remineralisation of the tooth. The limited increase in the water sorption, solubility, and viscosity is attributed to the ion release. Moreover, increased ion release can possibly enhance tooth remineralisation. Thus, the second null hypothesis stating that RBS incorporating BAG+MPC would not significantly affect the ion release related to tooth remineralisation is rejected. A limitation of this study is that physical and mechanical evaluations were not carried out after aging the sealant. Therefore, further study is needed to examine the long-term maintenance of these properties.

Another approach for caries inhibition is the repellence of bacteria attachment [10]. A nonspecific protein adsorption is a prerequisite for initial bacterial adhesion and subsequent biofilm formation [26,28]. In this regard, reducing protein adsorption is essential in bacteria attachment. In this study, protein-repellent properties of RBS were evaluated. BAG+MPC showed a significant difference from those of the control groups. Regarding the protein-repellent mechanism, when zwitterion forms the hydration shell with a large number of free water, it acts as an energy barrier that repels the nonspecific adsorption of proteins and bacteria [29,51]. Furthermore, bacterial attachment is a critical pathogenic event in biofilm formation since it represents a turning point for the planktonic bacteria, thereby leading to biofilm formation [43]. Dental plaque is a biofilm produced by a bacterial community composed of over 700 species [44]. One of the main components of this plaque is *S. mutans,* which is also considered as the major etiology of dental caries and plaque formation [28,36]. This study clearly demonstrated the significant antimicrobial effects of BAG+MPC against the attachment of *S. mutans* through FE-SEM images and WST assay.

The inhibition of biofilm accumulation is the most important strategy to overcome the drawbacks related to secondary caries [17]. Biofilm in the oral cavity is a complex system consisting of multi-species bacteria [17,22]. To confirm this property, we fabricated a saliva-derived biofilm model and evaluated the thickness and biomass of the formed multi-species biofilm by analysing the CLSM images. As secondary caries do not develop immediately after the sealant is applied to the pits and fissures, the bioactive properties of novel RBS are expected to have long-term durability. To evaluate its longevity, we used an artificial aging technique: static immersion aging and thermocycling aging [32]. There was no significant difference in the inhibitory effect between the non-aged and aged groups, which exhibited the sustained antibacterial effect, which is suitable for RBS. Therefore, our third null hypothesis stating that RBS incorporating BAG+MPC would not significantly alter the antimicrobial properties is also rejected. 

Although the limitation of this study was that the remineralisation and antimicrobial abilities of this novel material were not obviously evaluated, the possibility of these properties were clearly confirmed. To the best of our knowledge, this is the first study to evaluate the potential for remineralisation and antimicrobial capabilities of RBS incorporating BAG and MPC. Despite the limitations of this study, we confirmed that the novel RBS has the advantages of remineralisation and inhibition for bacteria in contrast to commercial RBS, consequently resulting in its resistance to secondary caries. 

## 5. Conclusions

The novel RBS incorporating BAG+MPC was investigated for the first time. It exhibited both enamel remineralisation and bacteria and biofilm inhibition properties. Furthermore, the inhibition of biofilm formation was sustained for a prolonged period. Therefore, RBS incorporating BAG and MPC is a promising candidate for dental materials in preventing secondary caries. 

## Figures and Tables

**Figure 1 nanomaterials-10-01581-f001:**
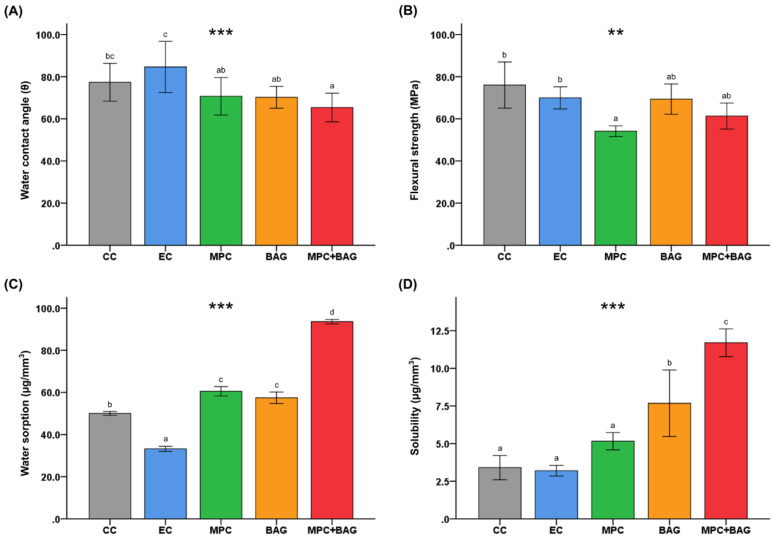
The water contact angle (**A**), flexural strength (**B**), water sorption (**C**), and water solubility (**D**) of the samples: Commercial control (CC), experimental control of 50 wt% resin matrix + 50 wt% glass filler (EC), 48.5 wt% resin matrix + 48.5 wt% glass filler +3 wt% MPC (MPC), 50 wt% resin matrix + 37.5 wt% glass filler + 12.5 wt% BAG (BAG), and 48.5 wt% resin matrix + 36.38 wt% glass filler + 12.12 wt% BAG + 3 wt% MPC (MPC+BAG). The different letters above the bars indicate the significant differences. *** *p* < 0.001, ** *p <* 0.01 for comparison between the groups.

**Figure 2 nanomaterials-10-01581-f002:**
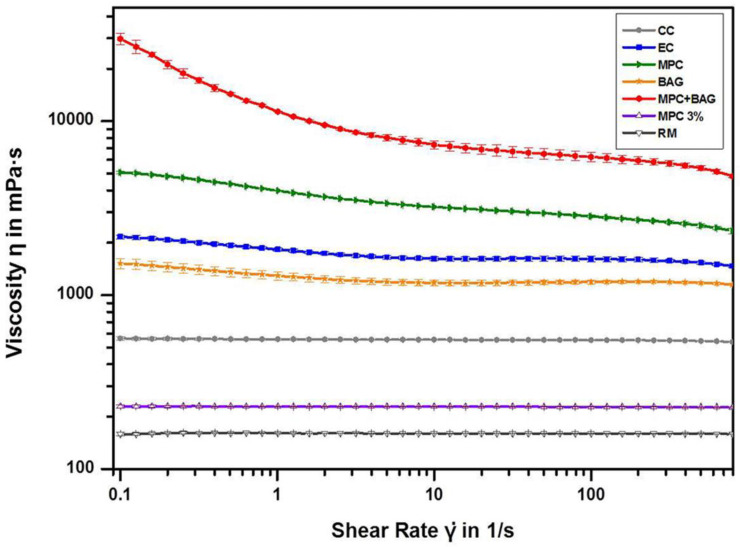
Viscosity of the groups: Commercial control (CC), experimental control of 50 wt% resin matrix + 50 wt% glass filler (EC), 48.5 wt% resin matrix + 48.5 wt% glass filler +3 wt% MPC (MPC), 50 wt% resin matrix + 37.5 wt% glass filler + 12.5 wt% BAG (BAG), 48.5 wt% resin matrix + 36.38 wt% glass filler + 12.12 wt% BAG + 3 wt% MPC (MPC+BAG), resin matrix without glass filler (RM), and resin matrix with MPC 3 wt% and without glass filler (MPC 3%).

**Figure 3 nanomaterials-10-01581-f003:**
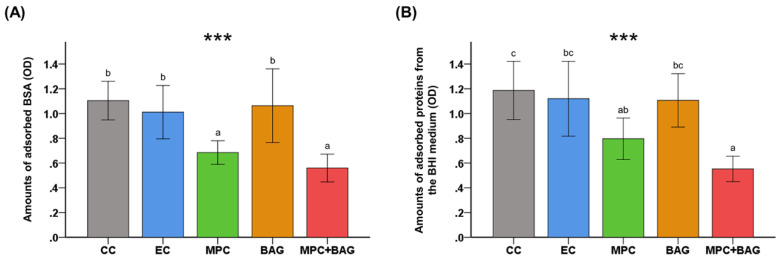
Comparison of the optical density (OD) of adsorbed bovine serum albumin (BSA) (**A**) and protein adsorbed form brain heart infusion (BHI) (**B**) between different groups: Commercial control (CC), experimental control of 50 wt% resin matrix + 50 wt% glass filler (EC), 48.5 wt% resin matrix + 48.5 wt% glass filler +3 wt% MPC (MPC), 50 wt% resin matrix + 37.5 wt% glass filler + 12.5 wt% BAG (BAG), and 48.5 wt% resin matrix + 36.38 wt% glass filler + 12.12 wt% BAG + 3 wt% MPC (MPC+BAG). Different letters above the bars indicate significant differences. *** *p* < 0.001 for comparison between all groups.

**Figure 4 nanomaterials-10-01581-f004:**
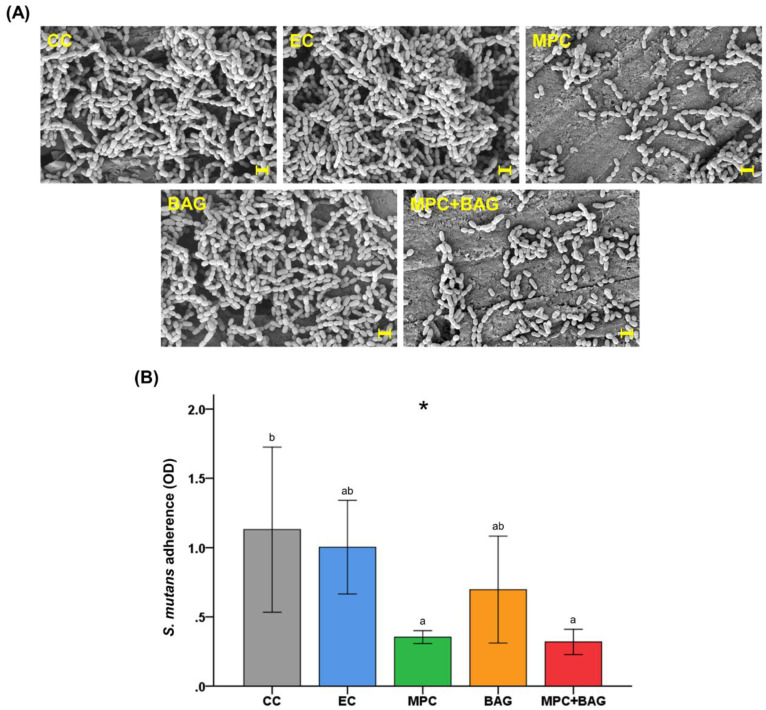
Representative scanning electron images of *S. mutans* attached to the surfaces of the groups at a magnification of 10,000× (**A**). The scale bar is 1 µm. Water-soluble tetrazolium counts derived from *S. mutans* attached on the surfaces of each group (**B**). The groups are: Commercial control (CC), experimental control of 50 wt% resin matrix + 50 wt% glass filler (EC), 48.5 wt% resin matrix + 48.5 wt% glass filler +3 wt% MPC (MPC), 50 wt% resin matrix + 37.5 wt% glass filler + 12.5 wt% BAG (BAG), and 48.5 wt% resin matrix + 36.38 wt% glass filler + 12.12 wt% BAG + 3 wt% MPC (MPC+BAG). The different letters above the bars indicate significant differences. * *p* < 0.05 for the comparison between all groups.

**Figure 5 nanomaterials-10-01581-f005:**
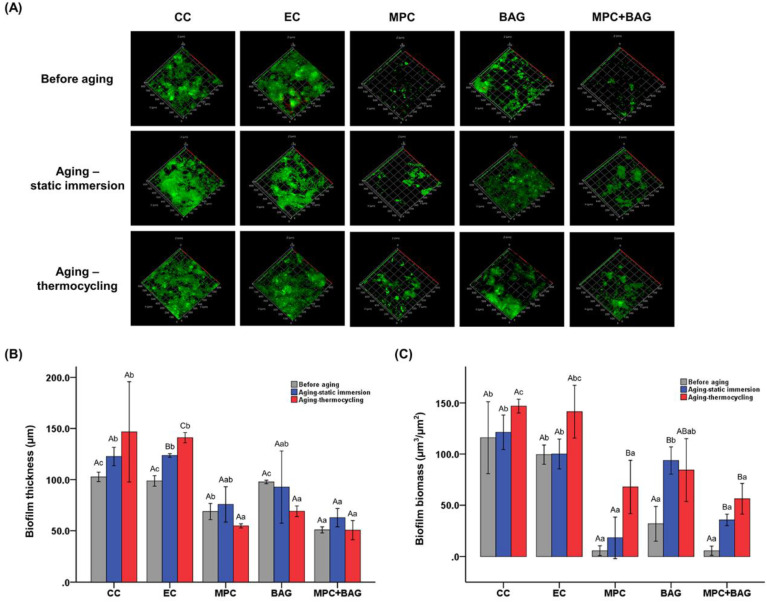
Representative live/dead staining images of the biofilm attached on the surfaces in the groups under three different conditions: before aging, with static immersion aging, and with thermocycling aging (**A**). Quantitative analysis of the thickness (**B**) and biomass (**C**) of the biofilms. The groups are: Commercial control (CC), experimental control of 50 wt% resin matrix + 50 wt% glass filler (EC), 48.5 wt% resin matrix + 48.5 wt% glass filler +3 wt% MPC (MPC), 50 wt% resin matrix + 37.5 wt% glass filler + 12.5 wt% BAG (BAG), and 48.5 wt% resin matrix + 36.38 wt% glass filler + 12.12 wt% BAG + 3 wt% MPC (MPC+BAG). The uppercase and lowercase letters indicate the statistically significant differences within the groups and between groups, respectively.

**Table 1 nanomaterials-10-01581-t001:** Released concentrations of Na^+^, Ba^2+^, Ca^2+^, Al^3+^, SiO_3_^2−^, BO_3_^3−^, and PO_4_^3−^ ions from the samples *.

	Released Concentration (ppm)										*p* Value
	CC		EC		MPC		BAG		MPC+BAG		
	Mean	SD	Mean	SD	Mean	SD	Mean	SD	Mean	SD	
Na^+^	0.23 ^a^	0.06	0.38 ^a^	0.10	0.49 ^a^	0.15	20.32 ^b^	3.78	19.71 ^b^	0.99	<0.001
Ba^2+^	0.14	0.18	0.44	0.13	0.39	0.20	0.65	0.84	1.66	1.43	0.201
Ca^2+^	0.25 ^a^	0.01	0.31 ^a^	0.03	0.26 ^a^	0.00	1.60 ^b^	0.18	1.99 ^c^	0.05	<0.001
Al^3+^	0.02 ^a^	0.00	0.01 ^a^	0.01	0.03 ^a^	0.02	0.08 ^b^	0.01	0.12 ^b^	0.02	<0.001
SiO_3_^2−^	0.47 ^a^	0.09	0.43 ^a^	0.02	0.76 ^a^	0.16	6.15 ^b^	1.00	9.39 ^c^	0.87	<0.001
BO_3_^3−^	0.05 ^a^	0.01	0.90 ^c^	0.05	2.12 ^d^	0.06	0.57 ^b^	0.06	8.20 ^e^	0.18	<0.001
PO_4_^3−^	0.25 ^a^	0.43	0.00 ^a^	0.00	1.26 ^a^	0.61	1.19 ^a^	0.08	106.39 ^b^	5.35	<0.001

Different letters indicate statistically significant differences between groups. * Commercial control (CC); Experimental control of 50 wt% resin matrix + 50 wt% glass filler (EC); 48.5 wt% resin matrix + 48.5 wt% glass filler +3 wt% MPC (MPC); 50 wt% resin matrix + 37.5 wt% glass filler + 12.5 wt% BAG (BAG); 48.5 wt% resin matrix + 36.38 wt% glass filler + 12.12 wt% BAG + 3 wt% MPC (MPC+BAG).

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
