# Peer review of "Resin-Based Sealant with Bioactive Glass and Zwitterionic Material for Remineralisation and Multi-Species Biofilm Inhibition"

_nanomaterials, 2020, doi:10.3390/nano10081581_

Round 1
Reviewer 1 Report
Antibiofilm activity of MPC has been well studied and reported in various dental materials including resin based system (https://patentimages.storage.googleapis.com/bd/e0/f2/6346f2fff1bfb1/US20170202752A1.pdf, A novel protein-repellent dental composite containing 2-methacryloyloxyethyl phosphorylcholine, International Journal of Oral Science volume 7, pages103–109(2015)).
The combination of Bioactive glass and MPC has also been studied and reported in dental materials, (Synergetic Effect of 2-Methacryloyloxyethyl Phosphorylcholine and Mesoporous Bioactive Glass Nanoparticles on Antibacterial and Anti-Demineralisation Properties in Orthodontic Bonding Agents (https://www.mdpi.com/2079-4991/10/7/1282/htm) even in resin based systems (Bioactive resin-based composite with surface pre-reacted glass-ionomer filler and zwitterionic material to prevent the formation of multi-species biofilm (https://www.sciencedirect.com/science/article/pii/S0109564119302118)
This study has rather limited level of novelty, just introducing known components into a system; subsequent findings of remineralizing effect and antibiofilm properties are obvious for the person skilled in the art.
Specific details with regard to the manuscript
1- Fissure sealants are used to prevent caries in pit and fissures on the occlusal surface. Usually they are suitable for young age 6-16 yrs children. To prevent primary caries if used for prevention, hence it is not a 100% tooth filling material. These techniques are less applicable for adults.
2- The addition of BAG has been a gold standard in filling and bone regeneration biomaterials. However, adding BAG to fissure sealant would cumbersome sealant usability given its effect on flow and applied layer thickness.
The authors have added BAG as source of minerals, I couldn’t find particle size. It is important to know particle size they used in this study. As fissure sealant has to be designed to be applied as a thin layer to cover the fissure and fill up the micro-irregularities. Using large particle BAG would render its less useful and less practical. Yet upon releasing the ions, i.e. dissolving BAG this would result in voids inside the material weakening its properties. The authors have not studied material's physical properties after ageing
3- The whole bacterial attachment section is needs more details. Only one bacterial species has been tested for attachment and viability . It would’ve been great if they use CFU to culture bacteria and know the exact effect of the material on viability. The authors did not provide full details about initial bacterial concentration, incubation conditions, sample position in the incubation plate
4- The authors model is not a full model, but rather incubating saliva from donors onto the sample. There is no simulated environment for oral cavity, nor saliva flow furthermore pH change has not been representative. Also, more details about the used ageing technique.
The authors drew two solid conclusions
1- The material exhibit enamel re-miniralisation
The authors did not evaluate reminierlisation but only measured ion release. It is less likely to draw such a solid conclusion from only knowing released ion concentration
2- The material is antimicrobial
The antibacterial tests used are not comprehensive and can not draw strong conclusion to be antibacterial material but has the potential to be antibacterial
A full panel of antimicrobial tests has to be accomplished before concluding a material to be antimicrobial. The authors did not measure minimum inhibitory concentration MIC, minimum bactericidal concentration MBC, disc diffusion, Colony forming unit CFU.
Relying on qualitative SEM on very small representative area with WST assay (colorimetric test) only on one bacterial species in aerobic environment. Also live/dead assay on 48hrs old biofilm. Would not fully back the claim of an antimicrobial material, but rather a material has the potential to inhibit bacterial growth, attachment and biofilm formation.
Author Response
|
Reviewer #1 |
|
|
General comment |
Antibiofilm activity of MPC has been well studied and reported in various dental materials including resin based system (https://patentimages.storage.googleapis.com/bd/e0/f2/6346f2fff1bfb1/US20170202752A1.pdf, A novel protein-repellent dental composite containing 2-methacryloyloxyethyl phosphorylcholine, International Journal of Oral Science volume 7, pages103–109(2015)). The combination of Bioactive glass and MPC has also been studied and reported in dental materials, (Synergetic Effect of 2-Methacryloyloxyethyl Phosphorylcholine and Mesoporous Bioactive Glass Nanoparticles on Antibacterial and Anti-Demineralisation Properties in Orthodontic Bonding Agents (https://www.mdpi.com/2079-4991/10/7/1282/htm) even in resin based systems (Bioactive resin-based composite with surface pre-reacted glass-ionomer filler and zwitterionic material to prevent the formation of multi-species biofilm (https://www.sciencedirect.com/science/article/pii/S0109564119302118) This study has rather limited level of novelty, just introducing known components into a system; subsequent findings of remineralizing effect and antibiofilm properties are obvious for the person skilled in the art. |
|
Response to General Comment |
First of all, thank you very much for your comments. We agree on the fact that the similar concept has been reported by several publications, but this manuscript is novel as we attempted to incorporate zwitterionic materials in resin-based sealant with bioactive glasses. They would have completely different requirements related to clinical applications such as viscosity, which is an important requirement for sealants. Although there was a limitation in increasing the viscosity in the results of the experiment, it is thought that p-ion plays a role in increasing the viscosity. Therefore, we plan to investigate the viscosity change according to the p-ion content in the future study. We now have added more details as follows to highlight this novelty: Also, viscosity is an important factor in the application of sealants. However, to date, there is yet to be a study on the viscosity and benefits of RBS containing BAG and MPC in providing both remineralisation and antibacterial capabilities. |
|
|
|
|
Comment 1 |
Fissure sealants are used to prevent caries in pit and fissures on the occlusal surface. Usually they are suitable for young age 6-16 yrs children. To prevent primary caries if used for prevention, hence it is not a 100% tooth filling material. These techniques are less applicable for adults. |
|
Response to Comment 1 |
Thank you for your valuable comments. I fully agree with your opinion that fissure sealants are not a 100% filling material, but is seems that the sealant could be used for preventing primary caries as well as type C of preventive resin restoration. I also agree that the technique is useful for children with age between 6 and 16. Details are now added in Introduction as; To inhibit this problem, pit and fissure sealants have been widely used as a preventive strategy, especially in children with age between 6 and 16 [6]. |
|
|
|
|
Comment 2 |
The addition of BAG has been a gold standard in filling and bone regeneration biomaterials. However, adding BAG to fissure sealant would cumbersome sealant usability given its effect on flow and applied layer thickness. The authors have added BAG as source of minerals, I couldn’t find particle size. It is important to know particle size they used in this study. As fissure sealant has to be designed to be applied as a thin layer to cover the fissure and fill up the micro-irregularities. Using large particle BAG would render its less useful and less practical. Yet upon releasing the ions, i.e. dissolving BAG this would result in voids inside the material weakening its properties. The authors have not studied material's physical properties after ageing |
|
Response to Comment 2 |
Thank you for your detailed review and we do apologize for lack of information related to BAG. According your suggestion, the ‘2. Materials and methods’, under ‘2.1 Preparation of 45S5 BAG and MPC powder’ and’2.2 Fabrication of novel RBS’, we now have include the details as follows: The ground powder was filtered through a 500-mesh sieve to obtain fine particles less than 25 µm size [21]. Silanized dental glasspowder was 180 ± 30 nm. Additionally, we agree with the opinion that the evaluation of features including some of physical properties after aging was not conducted. In accordance with a previous clinical study, pit and fissure sealants containing BAG had complete retention on the occlusal surface even after one year. [E Kouzmina, T Smirnova, N Pazdnikova: A one year clinical study of the efficacy of a pit and fissure sealant containing bioactive glass. Oral Health and Dental Management in the Black Sea Countries. 2009]. The relevant information has been added in the limitation of this study as follows: There is a limitation of this study that physical and mechanical evaluations were not carried out after aging the sealant. Therefore, further study is needed to examine the long-term maintenance of these properties. |
|
|
|
|
Comment 3 |
The whole bacterial attachment section is needs more details. Only one bacterial species has been tested for attachment and viability. It would’ve been great if they use CFU to culture bacteria and know the exact effect of the material on viability. The authors did not provide full details about initial bacterial concentration, incubation conditions, sample position in the incubation plate |
|
Response to Comment 3 |
Thank you for your comments. In the previous study, the results of viability using CFU and OD measurement were consistent. [MJ LEE, MK Kang: Analysis of the antimicrobial, cytotoxic, and antioxidant activities of cnidium officinale extracts. Plants. 2020], and in this study, the OD value of bacteria was measured using WST. Since this method has been verified in several studies, it is considered to be highly reliable. According to your suggestion, we will measure not only the CFU but also the viability of the bacteria in further study. Also, according to your comment, we’ve added details as follows: Following the preparation of disc-shaped specimens, 1 mL of bacterial suspension (1 × 108 cells/mL) was placed on each disc in a 24-well plate and incubated at 37 °C for 24 h. After incubation, the samples were gently washed twice with PBS to remove any non-adherent bacteria [17]. |
|
|
|
|
Comment 4 |
The authors model is not a full model, but rather incubating saliva from donors onto the sample. There is no simulated environment for oral cavity, nor saliva flow furthermore pH change has not been representative. Also, more details about the used ageing technique. The authors drew two solid conclusions 1- The material exhibit enamel re-miniralisation The authors did not evaluate reminierlisation but only measured ion release. It is less likely to draw such a solid conclusion from only knowing released ion concentration 2- The material is antimicrobial The antibacterial tests used are not comprehensive and can not draw strong conclusion to be antibacterial material but has the potential to be antibacterial A full panel of antimicrobial tests has to be accomplished before concluding a material to be antimicrobial. The authors did not measure minimum inhibitory concentration MIC, minimum bactericidal concentration MBC, disc diffusion, Colony forming unit CFU. Relying on qualitative SEM on very small representative area with WST assay (colorimetric test) only on one bacterial species in aerobic environment. Also live/dead assay on 48hrs old biofilm. Would not fully back the claim of an antimicrobial material, but rather a material has the potential to inhibit bacterial growth, attachment and biofilm formation. |
|
Response to Comment 4 |
Thank you for kind help on reviewing this article. We agree with the opinion that current data set may not be enough to assess remineralisation and antimicrobial effects. The focus of the study was to consider incorporating zwitterionic into resin-based sealants with bioactive glass, in order to assess potential of being those properties while maintaining original features that would be suitable for clinical application as sealants. Hence, our mineralization study was to consider ‘potential’ and therefore only with the concentration of released ions were considered. We are planning to evaluate the remineralisation that may be linked to the concentration of ions in ex vivo or other related study in our future planned study. Also, as the previous study demonstrated full set of antimicrobial test with MPC, our study was to just consider potential for anti-propellent properties, not bactericidal effects. According to your suggestion, we now have revised the manuscript as follows: Although the limitation of this study was that the remineralisation and antimicrobial ability of this novel material was not obviously evaluated, the possibility of these properties were clearly confirmed. To the best our knowledge, this is the first study to evaluate the potential for remineralisation and antimicrobial capabilities of RBS incorporating BAG and MPC. |
Reviewer 2 Report
This paper describes a series of experiments to test (in vitro) novel materials with enhanced properties to both contribute to the prevention of enamel lesions and also to enhance remineralisation should it occur.
The theoretical approach is very interesting. The experimental approach was comprehensive and provides convincing evidence that the research team have developed a material which deserves further scrutiny.
Overall the paper is very well written and straightforward to follow. There is sufficient detail for others to replicate and the data presented is similarly clear and all figures stand alone when accompanied by the legends.
The discussion section could be improved with minor changes.
These are:
Paragraph 1. after 'polymerisation' do you need to add 'shrinkage'? and ...can eventually lead to .... - it is not inevitable
Page 11, 5th paragraph: the sentence beginning..as long as RBS can be strongly attached to the tooth...it cannot prevent microleakage....
I think this needs reworking. I believe the authors are trying to say that as long if there is good attachment then microleakage will be minimised....
Thinking of the issue of applying this novel material to a tooth surface, the viscosity is critical. Successful fissure sealants can flow into pits and fissures with little encouragement leaving a smooth surface requiring little adjustment. I think some comment to link the increased viscosity and the clinical situation would be pertinent.
an example of this is the use of glass ionomer as a sealant. It does not flow easily, but can be placed by compressing a small amount onto the tooth surface and then curing it.
The conclusions are appropriate and supported by the evidence presented.
Author Response
|
Comment 1 |
Paragraph 1. after 'polymerisation' do you need to add 'shrinkage'? and ...can eventually lead to .... - it is not inevitable |
|
Response to Comment 1 |
Thank you for your helpful comments. According to your comment, we have modified sentence as follows: Particularly, marginal caries are formed around the sealed sealant on the material–tooth interfaces due to either the partial loss of materials or the microleakage and gaps induced by polymerisation shrinkage [13,14]. |
|
|
|
|
Comment 2 |
Page 11, 5th paragraph: the sentence beginning..as long as RBS can be strongly attached to the tooth...it cannot prevent microleakage.... I think this needs reworking. I believe the authors are trying to say that as long if there is good attachment then microleakage will be minimised.... |
|
Response to Comment 2 |
Thank you for specific comment and we are sorry for making a minor mistake. According to your suggestion, we’ve corrected the sentence as follow: As long as RBS can be strongly attached to the tooth, it can prevent microleakage at the tooth–RBS margin [4] |
|
|
|
|
Comment 3 |
Thinking of the issue of applying this novel material to a tooth surface, the viscosity is critical. Successful fissure sealants can flow into pits and fissures with little encouragement leaving a smooth surface requiring little adjustment. I think some comment to link the increased viscosity and the clinical situation would be pertinent. an example of this is the use of glass ionomer as a sealant. It does not flow easily, but can be placed by compressing a small amount onto the tooth surface and then curing it. The conclusions are appropriate and supported by the evidence presented. |
|
Response to Comment 3 |
Thank you very much for your valuable comments. According to your recommendation, we have revised the manuscript as follows:
Although high viscosity glass ionomer, as a pit and fissure sealant, does not flow easily, it can be placed by curing after compressing a small amount on the tooth surface, which has been used effectively for a long time. |